# *Phf8* histone demethylase deficiency causes cognitive impairments through the mTOR pathway

Xuemei Chen[1,2,3], Shuai Wang [1], Ying Zhou[1], Yanfei Han[1,4], Shengtian Li [1], Qing Xu[2], Longyong Xu[2], Ziqi Zhu[2], Youming Deng[2], Lu Yu[2], Lulu Song[1], Adele Pin Chen[2], Juan Song[5], Eiki Takahashi[6], Guang He[1], Lin He[1], Weidong Li[1] & Charlie Degui Chen[2]

Epigenomic abnormalities caused by genetic mutation in epigenetic regulators can result in neurodevelopmental disorders, deficiency in neural plasticity and mental retardation. As a histone demethylase, plant homeodomain finger protein 8 (*Phf8*) is a candidate gene for syndromal and non-specific forms of X-chromosome-linked intellectual disability (XLID). Here we report that *Phf8* knockout mice displayed impaired learning and memory, and impaired hippocampal long-term potentiation (LTP) without gross morphological defects. We also show that mTOR signaling pathway is hyperactive in hippocampus in *Phf8* knockout mouse. Mechanistically, we show that demethylation of H4K20me1 by *Phf8* results in transcriptional suppression of RSK1 and homeostasis of mTOR signaling. Pharmacological suppression of mTOR signaling with rapamycin in *Phf8* knockout mice recovers the weakened LTP and cognitive deficits. Together, our results indicate that loss of *Phf8* in animals causes deficient learning and memory by epigenetic disruption of mTOR signaling, and provides a potential therapeutic drug target to treat XLID.

[1] Bio-X Institutes, Key Laboratory for the Genetics of Development and Neuropsychiatric Disorders (Ministry of Education), Shanghai Key Laboratory of Psychotic Disorders, and Brain Science and Technology Research Center, Shanghai Jiao Tong University, 800 Dongchuan Road, Shanghai, 200240, China. [2] State Key Laboratory of Molecular Biology, Shanghai Key laboratory of Molecular Andrology, Institute of Biochemistry and Cell Biology, Shanghai Institutes for Biological Sciences, Chinese Academy of Sciences, Shanghai, 200031, China. [3] Department of Anesthesiology, Ren Ji Hospital, School of Medicine, Shanghai Jiao Tong University, Shanghai, 200127, China. [4] Discipline of Neuroscience and Department of Anatomy and Physiology, School of Medicine, Shanghai Jiao Tong University, Shanghai, 200025, China. [5] Department of Pharmacology and Neuroscience Center, University of North Carolina School of Medicine, Chapel Hill, NC 27514, USA. [6] Research Resources Center, RIKEN Brain Science Institute, 2-1 Hirosawa, Wako, Saitama 351-0198, Japan. Xuemei Chen, Shuai Wang, Ying Zhou, Yanfei Han and Shengtian Li contributed equally to this work. Weidong Li and Charlie Degui Chen jointly supervised this work. Correspondence and requests for materials should be addressed to W.L. (email: liwd@sjtu.edu.cn) or to C.D.C. (email: cdchen@sibcb.ac.cn)

I ntellectual disability is a heterogeneous neurodevelopmental disorder characterized by impaired intellectual and adaptive functioning[1]. Genetic deficiency in X chromosome has been identified as one of the most important causes of intellectual disability, based on the clinical observation that mental retardation occurs more often in males than in females[2]. Large-scale genetic analysis and functional studies have revealed the causal relationship between genetic mutations, deletions or duplications in X chromosome and X-linked intellectual disability (XLID)[2–7]. For example, mutation of *FMR1* gene at Xq27 leads to dendritic spine abnormalities, impaired synaptic plasticity and severe mental retardation[8,9].

Systematic mutation screening of brain-expressed genes and linkage analysis of familial mental retardation have identified plant homeodomain finger protein 8 (*Phf8*, located in Xp11.2) as one of XLID-associated genes[10–12]. PHF8 contains two functional domains, an amino-terminal PHD finger recognizing lysine-methylated histones and a JmjC domain catalyzing lysine

demethylation. Previous in vitro demethyaltion assays have shown that PHF8 can demethylate histone H3K9me2/me1, H4K20me1, and/or H3K27me2[13–16]. Genetic silencing of *Phf8* in cultured cells leads to a delay in G1–S transition during cell cycle progression and impaired neuronal differentiation[14,17]. In vivo functional studies have revealed that loss of PHF8 causes apoptosis of neural cells in zebrafish and compromised locomotion in nematode, respectively[13,15]. Together, these studies provide evidence for the potential role of PHF8 in regulating cell differentiation and survival. However, the role of PHF8 in neural and cognitive function within mammalian brains remains unknown.

Homeostatic protein translation in neurons is critical for activity-dependent synaptic plasticity and cognitive function. Ribosomal S6 kinase (RSK) controls protein translation by promoting signaling cascade of mammalian target of rapamycin (mTOR), a serine/threonine kinase regulating translation rate and long-lasting synaptic plasticity[18,19]. Hyperactive mTOR signaling cascade and overactivation of local dendritic translation have

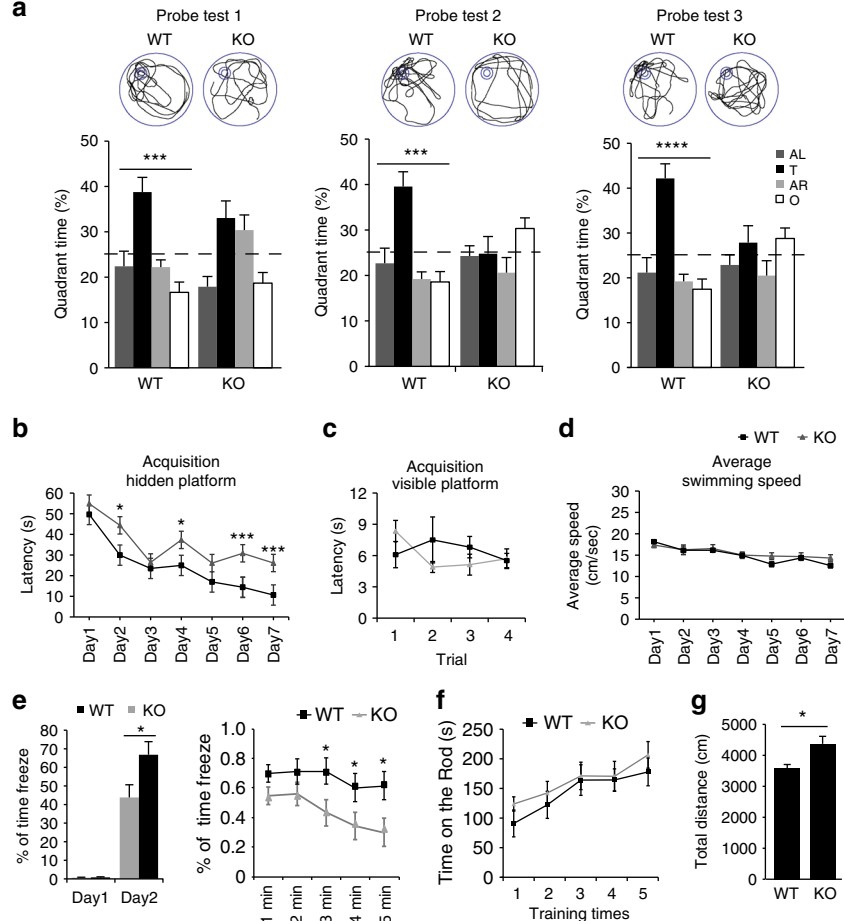

**Fig. 1** *Phf8* null mice show impaired learning and memory. **a** Morris water maze. Percentage time spent in each quadrant during the probe trials and representative swim trajectory of each group were given after 3 (probe test 1), 5 (probe test 2), and 7 (probe test 3) days of training (wild type (WT), $n = 17$; KO, $n = 11$; paired two-tailed *t*-test for water maze test). Four quadrants: adjacent left (AL), target quadrant (T), adjacent right (AR), opposite quadrant (O). **b** The escape latency time to reach the hidden platform during the 7-day training (WT, $n = 17$; KO, $n = 11$; unpaired two-tailed *t*-test). **c** The escape latency time to reach the visible platform within four trials given after probe test 3. **d** Average swimming speed of animals during the training with hidden platform. **e** Contextual fear conditioning test. Left, the baseline freezing time (Day 1) was recorded before foot shocks exposure to mice. The percentage of freezing time in the training context was measured 24 h after the conditioning (Day 2). Right, the freezing behavior of every minute observed during testing phase of contextual fear conditioning. The percentage of freezing time displayed significant difference in the last 3 min (WT, $n = 15$; KO, $n = 12$; unpaired two-tailed *t*-test). **f** Motor performance on the rota-rod. Both WT and Phf8 KO mice were given 5 trials on an accelerating rota-rod (4–40 r.p.m. over 5 min) within 1 day. The average duration time that mice stay on rota-rod in each trail was shown. (WT, $n = 10$; KO, $n = 10$). **g** Open-field test. The total moving distance traveled in 20 min. (WT, $n = 20$; KO, $n = 19$; unpaired two-tailed *t*-test,). All data are represented as mean ± s.e.m. *$p < 0.05$, ***$p < 0.001$ and ****$p < 0.0001$)

recently been reported in mouse models of tuberous sclerosis, Fragile X and Down's syndrome, all of which display intellectual disability[20,21].

Here, we show that *Phf8* knockout mice display impaired long-term potentiation (LTP) and deficiency in learning and memory. The epigenetic disruption of RSK-mTOR-S6K signaling is involved in cognitive defects by loss of *Phf8* and that the FDA-approved mTOR inhibitor rapamycin can rescue the behavioral and LTP deficits caused by *Phf8* deletion.

## Results

**Generation of *Phf8* null mice**. The lack of animal model with mutant *Phf8* impedes the progress in uncovering the cellular and molecular mechanisms underlying XLID. To model mental retardation in humans with PHF8 deficiency, we generated *Phf8* knockout (KO) mice by targeting the exons 7 and 8 encoding the core region of mice PHF8. The strategy of generating KO allele was described in detail in previous work[22]. Genotyping results showed recombinase-mediated efficient deletion of exons 7 and 8 in the genome of mutant mice (Supplementary Fig. 1a). The genetic disruption of *Phf8* was validated at protein level by immunoblotting (Supplementary Fig. 1b). We further confirmed the loss of *Phf8* in cortical and hippocampal neurons by immunostaining on *Phf8*-deficient brains (Supplementary Fig. 1c).

**_Phf8_ null mice show learning and memory impairment**. To address whether *Phf8* knockout mice mimic the intellectual disability in patients, we evaluated the learning and memory capacity of mice. Morris water maze was used to determine their spatial learning and memory. In the test, the control wild-type (WT) mice learned to use spatial cues to navigate a direct path to the hidden platform and displayed a significant preference for target quadrant at 3 (probe test 1), 5 (test 2), and 7 (test 3) days after training (Fig. 1a). However, the mutant mice exhibited significantly increased escape latency during the training process (Fig. 1b). In probe trials, we performed to assess spatial memory, the preference for target quadrant was strikingly compromised in KO mice as compared with WT mice (Fig. 1a). The observation that the swimming speed during training and the escape latency in water maze test with visible platform were comparable between control and mutant mice rules out the effect of motor and perceptual abilities on the spatial learning and memory (Fig. 1c, d).

We also evaluated whether PHF8 is essential for consolidation of fear memory with contextual fear conditioning test, and found that the freezing time of mutant mice receiving foot shock 1 day before test was shorter than that of control mice (Fig. 1e). However, additional behavioral tests showed that *Phf8* mutant mice did not display any motor disability but, instead, a slight increase in spontaneous activity (Fig. 1f, g). Taken together, these results suggest that learning and memory are impaired in *Phf8* null mice.

The knockout mice were born at the expected Mendelian ratio, and have similar weight and size to their wild-type littermates at both neonatal and adult stages (Supplementary Fig. 2a and b and Supplementary Fig. 3a and b). Although a subpopulation of XLID patients with PHF8 mutations display cleft lip and palate, we did not observe any deficits in craniofacial and limb development (Supplementary Fig. 3c and d). These results suggest that *Phf8* is dispensable for gross development of mice.

**_Phf8_ null mice show LTP deficit**. As the most well-characterized form of synaptic plasticity, activity-dependent long-term potentiation (LTP) in the hippocampus has long been considered as a cellular mechanism underlying learning and memory[23–25]. We next focused on hippocampal-dependent spatial learning and

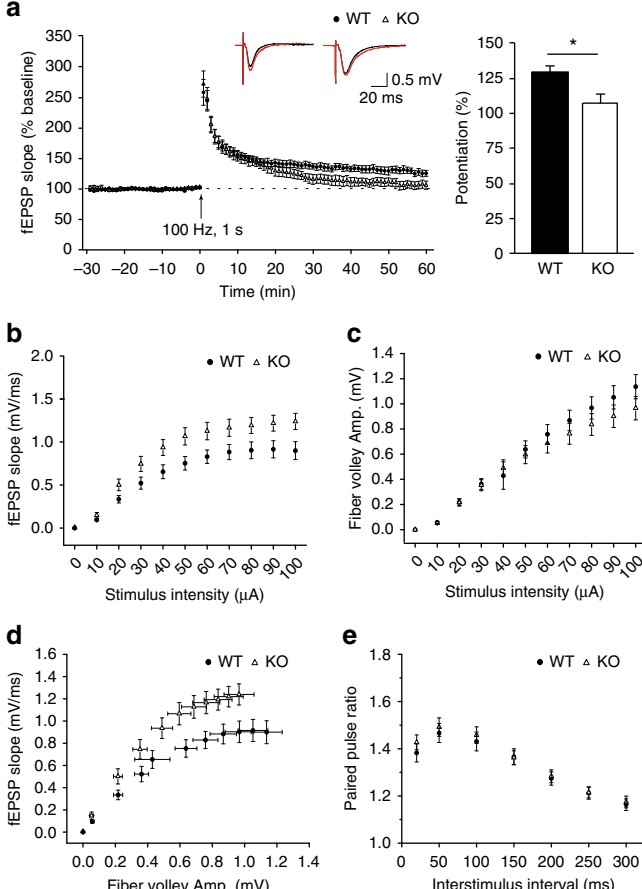

**Fig. 2** *Phf8* null mice display compromised long-term potentiation and enhanced basal synaptic transmission. **a** Shown are the slopes of field excitatory postsynaptic potentials (fEPSP) before and after tetanic stimulation (a train of 100 Hz stimulation for 1 s) recorded from hippocampal slices (WT, $n = 7$ slices from five mice; KO, $n = 8$ slices from six mice). Sample traces show averaged baseline responses (black) and responses during the last 10 min of recording (red) in WT and Phf8 KO animals. The ratio between fEPSP during the last 10 min recording and baseline is analyzed to evaluate the potentiation (right panel, $129.3 \pm 5.1\%$ in WT vs. $107.4 \pm 6.0\%$ in KO; unpaired two-tailed *t*-test, $p = 0.0164$). **b** Plot of input–output relationship (I–O curve) at Schaffer collateral-CA1 synapses in the hippocampal slices (one-way ANOVA, $F(1, 207) = 14.401$, $p = 0.0002$). The I–O curve obtained from PHF8-mutant mice is left-shifted compared with control (WT, $n = 10$ slices from five mice; KO, $n = 9$ slices from five mice). **c** Presynaptic fiber volley amplitude measured at increasing stimulus intensities (10–100 μA, one-way ANOVA, $F(1, 207) = 1.000$, $p = 0.3185$). **d** Scatter plot of fEPSP slope vs. diverse fiber volley amplitudes. The horizontal and vertical bars indicate the standard errors for fiber volley amplitudes and slope of the fEPSP, respectively. **e** Paired-pulse ratio measured at different inter-stimulus intervals reveals no differences in presynaptic release of transmitter between WT and Phf8 KO groups ($n = 10$ slices from five mice for each genotype; one-way ANOVA, $F(1,138) = 0.444$, $p = 0.5063$). All data are shown as mean ± s.e.m. *$p < 0.05$

memory by investigating the activity-dependent LTP in the hippocampus, and field excitatory postsynaptic potential (fEPSP) at Schaffer collateral-CA1 synapse was measured in acute hippocampal slices from control and *Phf8* null mice. As expected, we found that tetanic stimulation-induced LTP was compromised in *Phf8*-deficient hippocampus from adult mice (Fig. 2a). The basal synaptic transmission was also significantly increased, as shown by the higher value of the postsynaptic fEPSP slope against dose-

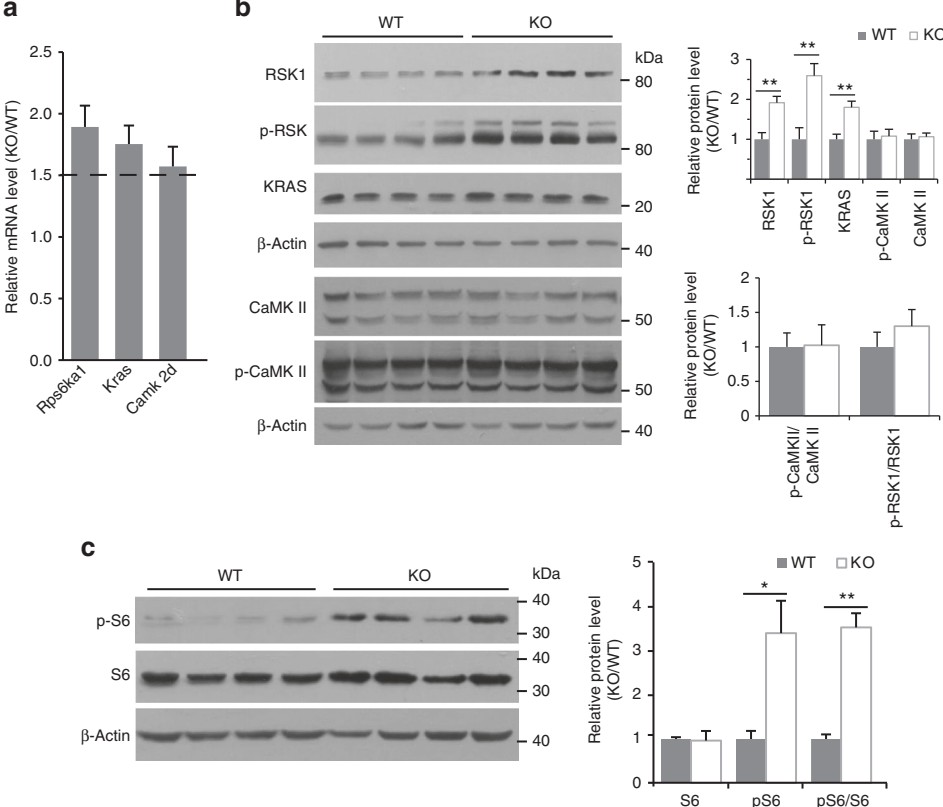

**Fig. 3** Loss of *Phf8* over activates mTOR signaling pathway in the hippocampus. **a** Shown is the relative mRNA expression of Neurotrophin and LTP associated genes in PHF8-deficient hippocampus, which is tested by real-time PCR and normalized to WT control ($n = 5$). The dotted line indicated that 1.5-fold change in mRNA level of KO mice compared to WT mice was used as a threshold. **b** Western blotting confirms the upregulation of RSK1, phosphorylated RSK1, and KRAS expression in *Phf8*-deficient hippocampus at the protein level. The statistical quantification is shown in the right panel ($n = 4$). **c** The hyperphosphorylation of ribosomal S6 in *Phf8* mutant mice is revealed by western blotting ($n = 4$). The statistical quantification is shown in the right panel. *$p < 0.05$ or **$p < 0.01$ vs. wild type

escalating intensities of stimuli (10–100 μA) in the KO mice (Fig. 2b).

To identify whether the enhancement of basal synaptic transmission in KO mice was mediated by pre- or postsynaptic mechanisms, we further analyzed changes in fiber volley amplitude at different stimulus intensities, as well as changes in the fEPSP slope at different fiber volley amplitudes. No difference was shown in the fiber volley amplitude vs. stimulus intensity plot between KO mice and WT mice (Fig. 2c). The input/output curves obtained by plotting the fEPSP slope vs. fiber volley amplitude were, however, significantly shifted to the left in KO mice compared with the curves obtained in the WT mice (Fig. 2d). In addition, paired-pulse facilitation (PPF) experiments across different inter-stimulus intervals revealed no differences between KO and WT mice (Fig. 2e). Taken together, these results suggest that postsynaptic changes rather than presynaptic changes occur in the *Phf8*-deficient mice.

***Phf8* null mice show hyperactive mTOR signaling**. To determine the molecular basis of defective LTP and learning/memory in *Phf8* mutant mice, we collected hippocampal tissues for gene expression profiling by microarray analysis. As compared with control mice tissues, we found that 1954 genes were upregulated while 1581 genes were downregulated with at least 1.5-fold change in PHF8-deficient hippocampus. One previous study identified genome-wide binding sites of PHF8 in human neuronal-like cell line by CHIP-seq (chromatin

immunoprecipitation followed by DNA sequencing)[15]. Thus, we converted these human homologous gene names into mouse using a table reported previously[26]. Among all the differential expressed genes in our data set, 526 upregulated and 471 downregulated genes contained PHF8-binding loci (Supplementary Fig. 4). Notably, *Rps6ka1*, *Kras* and *CamKII*, the key molecules in both Neurotrophin signaling pathway and LTP signaling pathway, were identified in gene ontology analysis in PHF8-deficient hippocampus (Supplementary Table 1 and Supplementary Table 2). Using quantitative real-time PCR, we confirmed the upregulation of transcripts in the KO mice (Fig. 3a). Concordantly, the protein expression of *Rps6ka1*-encoded RSK1 and *Kras*-encoded KRAS was significantly increased in the *Phf8* null hippocampus (Fig. 3b). We did not find the upregulation of total and phosphorylated CaMKII (including isoforms), although the mRNA level of *Camk2d* encoding CaMKIIδ was elevated in the mutant mice (Fig. 3a, b).

Signaling through Ras-RSKs-mTOR-S6K pathway regulates multiple processes involved in protein synthesis including translation initiation, rate, and biogenesis of ribosomes[27,28]. Upon stimulation of neurons with growth factors, RSKs are activated and phosphorylated by Ras-mitogen-activated protein kinase (MAPK) cascade[27,29]. The activation of RSKs proceeds to promote mTOR signaling to its downstream targets ribosomal S6 kinase (S6K) and then enhance protein translation[27]. Given that KRAS as one of Ras family members and RSK1 as one of RSK isoforms were upregulated by deletion of *Phf8* in hippocampus, we wondered whether RSKs-mTOR-S6K signaling cascade is

overactivated in mutant mice. Thus, we examined the phosphorylation level of RSK1, mTOR, and ribosomal protein S6 by immunoblotting. Indeed, loss of *Phf8* in the brain conspicuously increased phosphorylated RSK1, mTOR and S6 but not total expression of mTOR and S6 (Fig. 3b, c, Supplementary Fig. 5a and b), indicating hyperactive mTOR signaling in mutant mice.

**Phf8 regulates Rps6ka1 expression by H4K20me1 demethylation.** It has been suggested that H4K20me1, predominantly lying downstream from TSS, may participate in active transcription[30,31]. Recent studies identified PHF8 as an H4K20me1 demethylase and binds its regulated genes at the loci of transcription start site (TSS)[13,14], thus raising the question of whether PHF8 actively represses target gene expression by demethylation of H4K20me1. To investigate how PHF8 regulates gene expression, we performed Chip-qPCR as well as sequence assay and focused on four PHF8 targeting genes: *Rps6ka1*; *Kras*; *Camk2d*; and *Mapk14*, among which, *Rps6ka1, Kras, Camk2d* were overexpressed in *Phf8* KO mice, while *Mapk14* was not changed. Following immunoprecipitation with antibodies against PHF8, H3K9me2, or H4K20me1, we quantitatively analyzed the abundance of immunoprecipitated DNA fragments at the loci of transcription start site (TSS, amplicon 2), as well as upstream (amplicon 1) and downstream (amplicon 3) of TSS for each target gene (Fig. 4a). Our data showed that PHF8 specifically bound TSS of its target genes but not non-TSS sites (Fig. 4b). The genetic ablation of PHF8 abolished its binding to TSS, verifying the specificity of PHF8 antibodies for CHIP assay. Interestingly, loss of *Phf8* increased the enrichment of H4K20me1 rather than H3K9me2 at TSS of three overexpressed PHF8 targeting genes, but not at the TSS of *Mapk14*, whose expression was not changed in the hippocampus of *Phf8* KO mice (Fig. 4c, d and Supplementary Fig. 6). Our results from microarray and ChIP analyses coincide with previous observation showing that the magnitude of H4K20me1 peaks correlates with highly expressed genes[31,32]. Thus, these data suggest that PHF8 specifically binds to the TSS target genes including mTOR activator *Rps6ka1* and suppresses gene transcription by demethylating H4K20me1.

**mTOR inhibitor rescues learning/memory and LTP impairments.** Overactivation of mTOR signaling has recently been reported in mouse models with intellectual disability[20,21]. Suppression of hyperactive mTOR signaling by rapamycin rescues the long-term memory impairment in animal models with tuberous sclerosis or overexposure to cannabinoids[20,33]. Given the overactivation of Ras-RSKs-mTOR-S6K signaling cascade in *Phf8*-deficient mice, we sought to test whether rapamycin administration could ameliorate the learning and memory deficits of mutant mice. Because both hyperactive and hypoactive mTOR signaling impairs long-term synaptic plasticity in hippocampus[33,34], we first optimized the dose of rapamycin by evaluating the level of phosphorylated S6 (pS6) in hippocampus after drug application in animals for 3 consecutive days. Our data showed that low dose of rapamycin (5 mg kg$^{-1}$) rescued the pS6 to normal level but high dose (10 mg kg$^{-1}$) completely disrupted mTOR signaling (Supplementary Fig. 7). We then assessed the learning and memory performance of *Phf8* mutant mice in Morris water maze and Barnes maze after a 3-day pre-application of low-dose rapamycin. In water maze test, we performed the same procedure as previous two groups test (WT vs *Phf8* KO) that probe tests did at 3, 5, and 7 days following training. Importantly, rapamycin treatment improved the preference of mutant mice for target quadrant at day 5 in Morris water maze, although there were no differences between vehicle and rapamycin-treated groups at day 3 and 7 (Fig. 5a). The escape

latency to reach the hidden platform and the average swimming speed during training were also measured (Supplementary Fig. 8a and b). Barnes maze, another task to assess spatial memory, was also performed in our mice model. The data showed that rapamycin treatment improved the *Phf8* KO mice performance in the latency to the target hole during the training days (Fig. 5b) and numbers of target pokes (Fig. 5c), time in the zone (Supplementary Fig. 8g) in the probe test. Except for spatial memory, aversive memory was used to evaluate by contextual fear conditioning and passive avoidance test. We did not find the reversal of impaired cued memory (Supplementary Fig. 8c). However, *Phf8*-deficient mice with rapamycin treatment could enhance the fear memory in passive avoidance test (Fig. 5d). In addition, other locomotion and adaptive behavior were also assessed by rapamycin treatment. The mobility and social ability of mutant mice was not affected by rapamycin administration (Supplementary Fig. 8d and e). The increased spontaneous activity of mutant mice could not be rescued by rapamycin in *Phf8*-deficient mice (Supplementary Fig. 8f). Thus, we show that low-dose rapamycin ameliorates spatial memory loss in *Phf8* null mice.

Furthermore, we examined the effect of rapamycin treatment on abnormal synaptic transmission and plasticity in *Phf8* null hippocampus. The animals were treated with low-dose rapamycin for 3 days before and on the day of the electrophysiological recording on acute hippocampal slices (see Methods in detail). The results from extracellular field recording showed that rapamycin treatment significantly reversed the impaired LTP induction and shifted synaptic input–output curve in *Phf8*-deficient hippocampus in comparison to the control hippocampus (Fig. 6a–d). The paired-pulse facilitation ratio was not altered by rapamycin in either control or mutant mice (Fig. 6e). These findings indicate that application of rapamycin can rescue the abnormal synaptic transmission and plasticity, suggesting that overactivation of mTOR signaling pathway is involved in impaired learning/memory and synaptic potentiation.

## Discussion

Our results reveal that histone demethylase PHF8 epigenetically represses the transcriptional activation of Ras- and RSK1-encoding genes by demethylating H4K20me1 at TSS of target genes (Fig. 6f). The loss of *Phf8* in mammalian brain results in hyperactive mTOR signaling, impaired LTP, and learning/memory deficits. Administration of low-dose rapamycin to adult *Phf8*-deficient mice normalizes RSK-mTOR-S6K signaling pathway, and restores LTP and memory loss. This study uncovers the molecular and cellular mechanisms underlying *Phf8* mutation-mediated mental retardation, thus providing potential therapeutic strategy against XLID.

Aberrant epigenetic machinery caused by genetic mutations in one of the essential components has been shown to lead to X-linked mental retardation[9]. Large-scale genetic studies have identified a plethora of XLID-associated epigenetic regulators in X chromosome[2]. The epigenetic function of these genes such as *Fmr1, Mecp2, Ftsj1, Xnp, Jar1d1c, Znf41, Phf6,* and *Phf8* range from mRNA translation, DNA methylation, histone modification to chromatin remodeling[2]. Previous genetic studies demonstrated that patients with familial *Phf8* mutations display syndromic XLID with cleft lip and palate[10–12]. Here we generated *Phf8* knockout mice to model XLID in patients. The loss of *Phf8* in mouse brain caused significant deficiency in learning/memory and contextual cued memory. Interestingly, hippocampal activity-dependent LTP, one of the most important forms of synaptic plasticity underpinning learning and memory, was compromised in *Phf8* null mice. Beyond intellectual disability, impaired adaptive functioning is also observed in XLID patients. Social

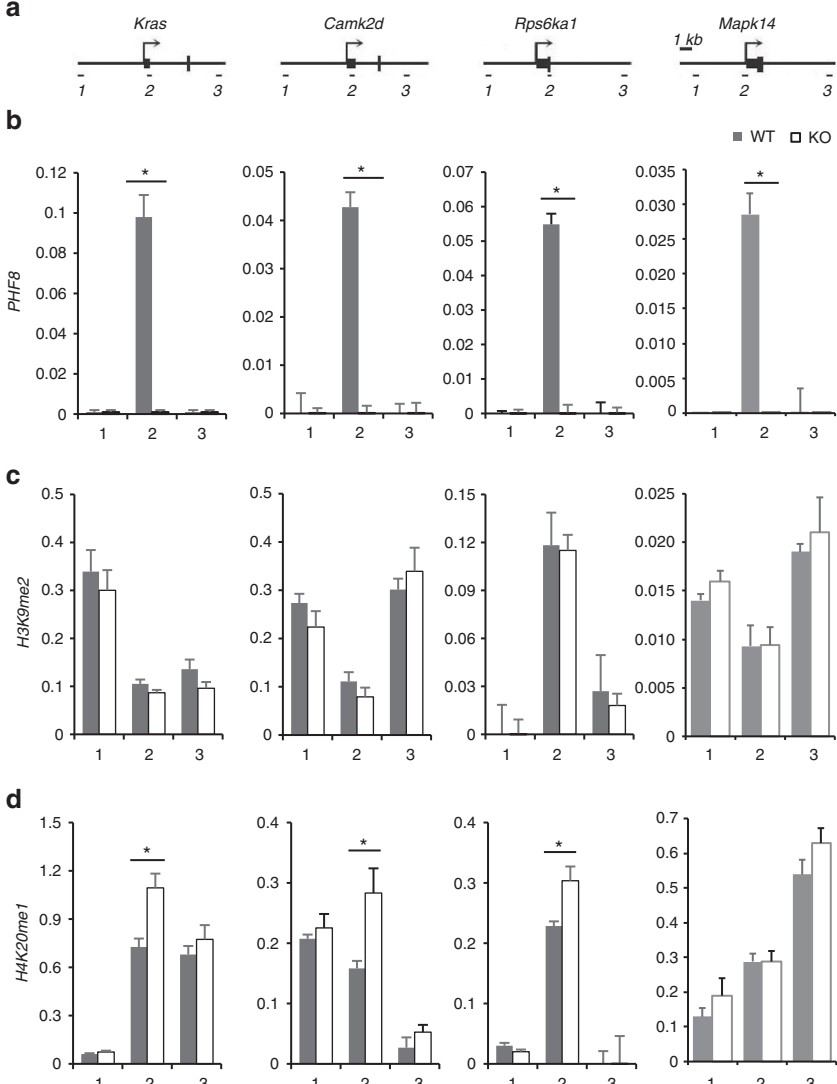

**Fig. 4** Depletion of PHF8 increases the enrichment of H4K20me1 on overexpressed genes in *Phf8* deficits. **a** Amplicons for real-time PCR are illustrated for three direct PHF8 target genes: *Kras*; *Camk2d*; *Rps6ka1*; and *Mapk14*. The black arrow represents the transcription start site (TSS) of genes, small black squares indicate exon. Short lines below represent the locations of the amplicons by different primers. Scale bar = 1 kb; **b** Conventional CHIP analysis with PHF8 antibody shows the specific binding of PHF8 on TSS loci of its targeting genes. **c** The enrichment of H3K9me2 on the PHF8 target genes is not changed by deletion of *Phf8*. **d** The enrichment of H4K20me1 on the overexpressed *Kras, Camk2d,* and *Rps6ka1* were enhanced by deletion of *Phf8*, while the enrichment of H4K20me1 was not changed in the TSS of *Mapk14*. Real-time PCR was performed on the input and immunoprecipitated genomic fragments. Data are presented as percentage of input. Numbers on the *x* axes represent the different amplicons. All data are shown as mean ± s.e.m. *$p < 0.05$; **$p < 0.01$ and ***$p < 0.001$

communication tests and behaviors in response to the environment are used to evaluate adaptive functions in mouse model studies, *Phf8* KO mice showed no significant difference in the social ability test here. However, *Phf8* KO mice display hyperactivity in the unfamiliar environment of open field and impairments in passive avoidance test and fear conditioning. In these unavoidable and unexpected conditions, mice need adaptive functions to deal with the stress and tasks. The role of PHF8 in adaptive behavior requires further investigations for making conclusions as we did for learning and memory here.

A recent paper reported that a different strain of *Phf8*-deficient mice showed resilience to stress induced anxiety- and depression-related behavior and no intellectual disability[35]. Although both our study and those in ref. [35] found the *Phf8*-deficient mice showed hyperactivity in open-field test, our *Phf8* KO mice displayed a significant intellectual disability in water maze, Barnes

maze and passive avoidance performance as well as LTP deficit. The difference of behavior phenotype in *Phf8*-deficient mice in two studies may be due to the genetic factors. Our testing mice were at least the fifth-generation progeny of a backcross into C57BL/6 J mice while the KO mice in ref. [35] were maintained on a 129/B6 background. In addition, subtle difference in the methods for behavioral testing may contribute to the difference in phenotype. Our study used a classical 5-min training protocol for contextual fear conditioning to evaluate fear memory, and measured the freezing level for 5 min 24 h later. The other study[35] used different training regimen and testing protocol which may have resulted in memory traces of different strength.

It has been suggested that homeostatic protein translation is precisely controlled by mTOR signaling during learning and memory processes[20]. Previous studies showed that disruption in mTOR pathway impairs learning/memory and synaptic

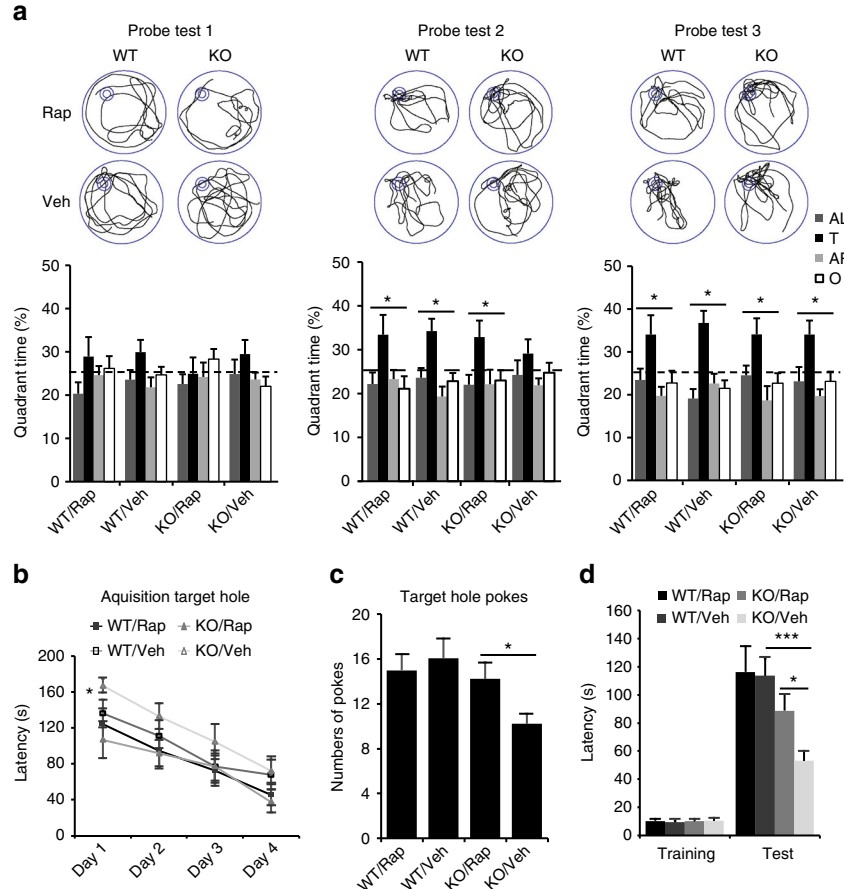

**Fig. 5** Rapamycin treatment rescues behavioral deficits in *Phf8* null mice. **a** Percentage time spent in each quadrant during the probe trials and representative swim trajectory of each group are evaluated in WT and *Phf8* KO mice with vehicle or low-dose rapamycin treatment. (WT/Veh, $n = 12$; WT/Rap, $n = 11$; KO/Veh, $n = 12$; KO/Rap, $n = 11$; paired two-tailed *t*-test, *$p < 0.05$). (Veh, vehicle injection; Rap, rapamycin treatment). **b** In Barnes maze, latency to reach the target hole is shown during training days. (WT/Rap, $n = 11$; WT/Veh, $n = 12$; KO/Rap, $n = 10$; KO/Veh, $n = 10$; unpaired two-tailed *t*-test, *$p < 0.05$). **c** Probe trial on day 5 is used to assess spatial memory retention. Number of pokes in target hole during the probe trial (90 s) are calculated and averaged (WT/Rap, $n = 11$; WT/Veh, $n = 12$; KO/Rap, $n = 10$; KO/Veh, $n = 10$; unpaired two-tailed *t*-test, *$p < 0.05$). **d** *Phf8* KO mice show deficit in the passive avoidance test and rapamycin treatment rescues the deficit. The bars indicate the mean latencies to enter the dark chamber on the training day (left) and 24 h later on the retention day (right). (WT/Rap, $n = 9$; WT/Veh, $n = 13$; KO/Rap, $n = 9$; KO/Veh, $n = 10$; unpaired two-tailed *t*-test, *$p < 0.05$, ***$p < 0.001$)

plasticity[20,33,34,36]. Our results showed hyperactive of mTOR signaling in *Phf8* mutant mice. And pharmacological inhibition of hyperactive mTOR signaling with rapamycin rescued the impaired LTP and memory loss in *Phf8* mutant mice. Taking together, the data indicated that *Phf8*-mediated overactivation of mTOR signaling is essential for normal cognition and synaptic plasticity. Besides, Akt-mTOR pathway was reported to be regulated by PHF8 in cardiac system and rapamycin treatment could rescue the effects of PHF8 loss in neonatal rat ventricle myocytes[37].

Morris water maze was widely used to evaluate spatial learning and memory in rodent. Our data (Fig. 5a) indicated that neither *Phf8* mutant mice nor their wild-type littermates with vehicle treatments had been enough trained to find the platform on probe test 1. However, after two more days' training, vehicle-treated *Phf8* mutant mice still displayed obvious learning deficit compared to wild-type littermates on probe test 2, while rapamycin injection rescue the learning deficit. And with excessive two more days training, all groups learned the location of the platform at the probe test 3. We noticed that vehicle-treated *Phf8* mutant and wild-type mice did not show the exact same performance at probe test 1 and probe test 3 as those without treatments in Fig. 1a. It may be due to some unknown effects on

water maze performance by daily injection. According to previous studies, daily injection as a mild stress, which can increase the synthesis and release of hypothalamic corticotrophin-releasing factor[38], had effects on rodent behaviors such as elevated plus maze and open field[39,40]. The difference on water maze performance of mutant mice with or without daily injection was also seen in a previous study[20]. The mechanism of the puzzle needs to be investigated in the future study.

As a histone demethylase, PHF8 has recently been reported to function as an H4K20me1 and H3K9me2 demethylase[13,14]. We found that PHF8 was specifically recruited to the TSS of its target genes. Loss of PHF8 in mouse hippocampus led to an increased enrichment of H4K20me1 but not H3K9me2 at TSS region, and thereby an upregulation of target genes including RSK1 and KRAS. To date, the role of H4K20me1 in transcriptional regulation remains controversial. Recent CHIP-seq analyses in mammalian cells show that H4K20me1 correlates with highly expressed genes and associates with TSS regions[30–32,41], while other studies suggest that the binding of H4K20me1 to promoter region mediates the transcriptional repression[14,42]. Therefore, the genomic loci of H4K20me1 enrichment may determine its role in transcriptional activation or repression. Our results in mammalian brain suggest that PHF8 represses transcription of target

genes such as *Rps6ka1* by demethylating H4K20me1, supporting the findings showing H4K20me1 as a marker for gene transcription activation[43].

Taken together, we provide an animal model mimicking the mental retardation of patients with *Phf8* mutation to study the neuropathology of XLID and show that loss of PHF8 in animals causes deficient learning and memory by epigenetic disruption of mTOR signaling. Importantly, low-dose rapamycin can reverse the impaired memory loss and synaptic plasticity, providing a therapeutic drug target to treat *Phf8*-associated XLID. Future studies with brain-wide neural circuit mapping and genome-wide epigenetic regulation of genes in *Phf8* mutant mice will further elucidate the circuit mechanisms underlying disrupted cued memory and increased spontaneous activity as well as epigenetic mechanisms for general gene regulation.

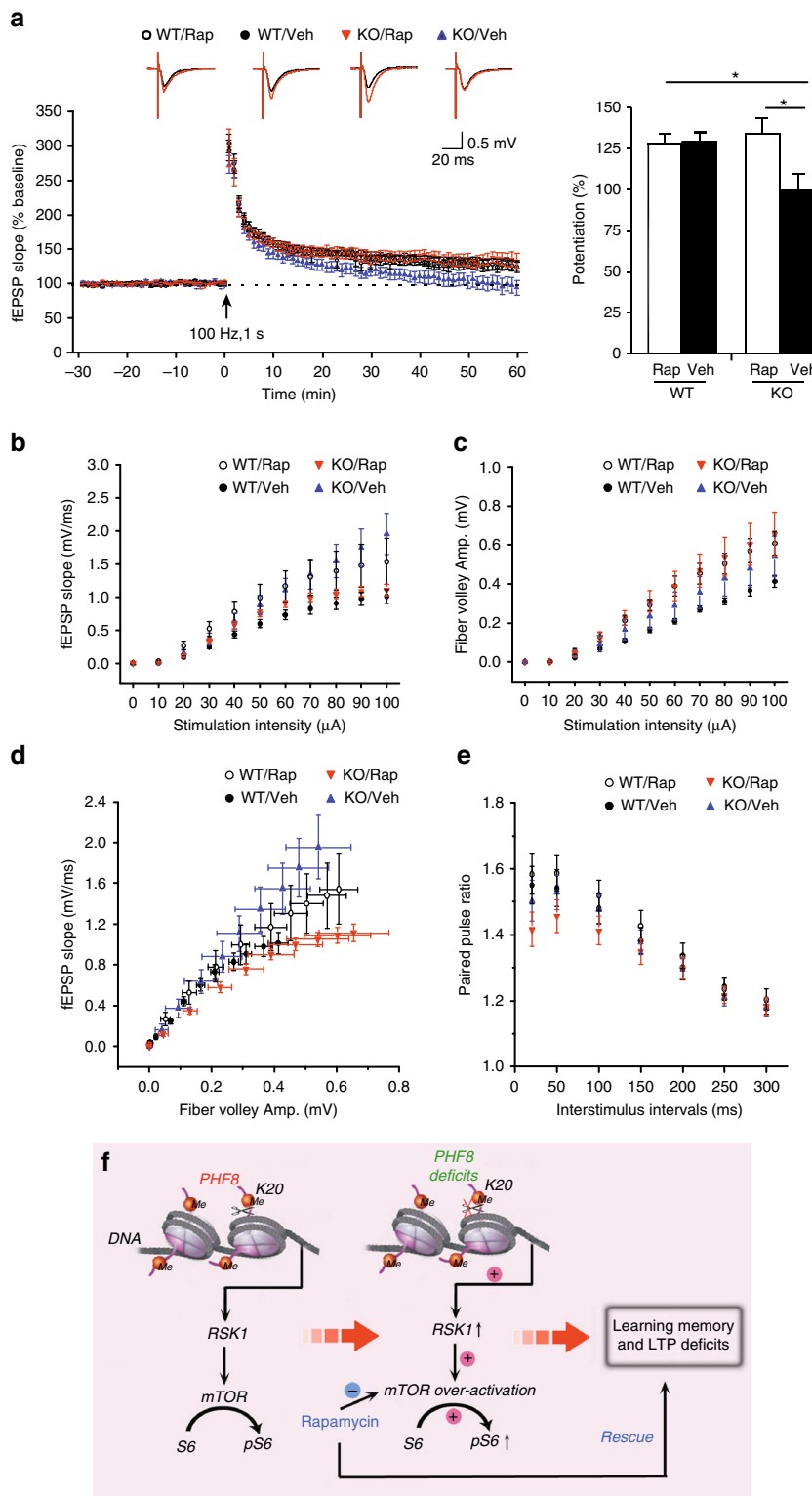

## Methods

**Mice**. To investigate the role of PHF8 in mammalian brain, we generated *Phf8* KO mice with a 129/S6/SvEv-background via homologous recombination. For the targeting vector, the exons 7 and 8 encoding the code region of the JmjC deme-thylase domain were flanked by two loxp sites. The strategy of design was presented in detail[22] in previous work. Embryonic stem (ES) cell candidates were screened via negative selection and genotyping. The heterozygous female *Phf8+/−* mice, generated by breeding *Phf8+/F* mice with *EIIa–Cre* mice, were backcrossed with C57BL/6 J wild-type male mice for at least 5 generations. The hemizygous male *Phf8−/Y* mice were then produced by crossing pure *Phf8+/−* heterozygotes with C57BL/6 J mice.

The mice were maintained on *ad libitum* food and water, and housed in groups of five to six mice per cage. The housing room was maintained at 23 °C on a 12 h light/dark. Eight to 12 weeks old male *Phf8−/Y* and their wild-type littermates were used in the study. For behavioral tests, mice were grouped by random double-blinded method. All animal procedures were conducted in accordance with the Guidelines for the Care and Use of Laboratory Animals and were approved by the Institutional Animal Care and Use Committee at Shanghai Institutes for Biological Sciences.

**Alcian blue and alizarin red staining**. Littermate animals obtained from each group were killed to collect the skin, muscle, and fat. The tissues were fixed in 100% ethanol for 4 days and then transferred to acetone for another 3 days. Mice were then stained with alcian blue (0.15%) in a solution containing ethanol and glacial acetic acid (8:2) for 2 days. Then, they were stained in an alizarin red (0.05%) solution containing 2% KOH. After staining, mouse samples were transferred to a 1% KOH solution until their soft tissues were dissolved, and they were then preserved in 100% glycerol.

**Water maze**. Water maze testing was conducted in a large circular pool (diameter: 120 cm, height: 50 cm) made of white plastic. The pool was filled to a depth of 35 cm with water (maintained at $20 \pm 1.0$ °C) covering a transparent circular platform 10 cm in diameter. The platform was submerged approximately 0.5–1.0 cm below the surface of the water. The pool was located in a room with diffuse lighting and with a number of visual cues hung on the wall and. The swimming activity of each mouse was monitored and tracked via a television camera mounted overhead, which relayed information, including variables such as latency to the platform, total distance traveled, time and distance spent in each quadrant, to a video tracking system.

Each day, a trial was initiated by placing each mouse in the water facing the pool wall in one of the four quadrants, which was randomly selected. For each training trial, the mouse was allowed to swim for 60 s to find the hidden platform. When successful, the mouse was allowed a 30 s rest period on the platform (timed manually with a stopwatch). If unsuccessful within the allotted time period, the mouse was guided to the platform and allowed a 30-s rest period on the platform. Each mouse underwent two session of two trails (inter-trial interval=30 s) for seven consecutive days. Two consecutive trails were considered to be one session. After all the animals had completed one session, they then performed the second session. The probe trials were conducted on days 3, 5, and 7 after training. The platform was removed from the pool and the mice were allowed to search for it for 60 s. The percentage of time spent in the target quadrant was analyzed. The swimming speed of tested mice below 10 cm s$^{-1}$ were regarded as floating and excluded from the data analysis.

**Contextual fear conditioning**. For contextual fear conditioning test was performed in the chambers from Med Associates Inc. During the training phase, mice were individually placed in the training context with three 0.75 mA foot shocks (2 s each, 1 min interval) after 2 min of exploration in the chambers. Training sessions were 5 min in duration. Mice were then returned to their home cages. 24 h after training, mice were placed back to the chamber with same training context for 5 min, the percentage of the freezing time was assessed for each mouse.

**Open field**. Mice were habituated in the testing room 60 min before the open-field test. Each mouse was gently placed in a square plexiglass box ((27.5 cm *L* × 27.5 cm *W* × 18 cm *H*) and was allowed to explore freely for 20 min. After each trial, the box was swept out with water that contained 0.1% detergent. The total distance of movement was automatically recorded and calculated using EthoVision XT v10.0 tracking software (Noldus, Wageningen, the Netherlands).

**Rota-rod test**. In the rota-rod test, mice were placed on an accelerating rota-rod cylinder (Ugo Basil, Italy). The rota-rod was accelerated from 4 to 40 r.p.m. in 5 min, the duration of each mice remained on the rota-rod during 5 min test. The time of duration was measured for five trails (inter-trial interval=60 min).

**Three-chamber social interaction test**. The Sociability test was measured in a three-chamber plexiglass box (60 cm *L* × 40 cm *W* × 30 cm *H*). Animals were habituated to the apparatus by placing them in the middle chamber and allowing them to explore the box for 10 min, with the doorways into the two side chambers open. In the test phase, an unfamiliar mouse (a 4-week-old C57BL/6 J female) was enclosed in a cylinder and placed in one of the side chambers, an empty cylinder placed in the other side chamber. The test mouse was allowed to explore the entire apparatus for a 5 min session. The duration time mouse stayed in every chamber and total sniffing time on cylinder were recorded by EthoVision XT tracking software.

**Passive avoidance performance**. Passive avoidance performance was assessed in Shuttle-box compartments (#ENV-010MC; Med Associates) in which one side of the shuttle-box was darkened and a light illuminated the other. On Day 1 (Training), each mouse was placed singly in the light chamber and allowed to explore by opening the door. When mouse entered into the dark chamber, the door controlled by the computer closed automatically. After 2 s entering the dark compartment, the mouse received a shock of 0.5 mA for maximum 5 s, and then mouse was returned to its home cage. The initial latency to enter the dark compartment was recorded. Animals remaining in the light compartment for more than 180 s during the training session received no shock and were excluded from the retention test. The memory retention test was performed on the next day without any shock; the mice were again positioned in the illuminated chamber and the time taken to enter the dark compartment was recorded as the retention latency. A maximum retention latency of 300 s was given to mice. After each trial the apparatus was cleaned.

**Barnes maze**. The Barnes maze consists of a circular platform (92 cm of diameter) with 20 equally spaced holes (5 cm diameter ; 2 cm around the hole define as zone; 7.5 cm between holes) along the perimeter and is elevated 105 cm above the floor. On the pre-training trial, the mouse was placed in the middle of the maze, and pre-trained to enter the escape box by guiding it to the escape box and remaining there for 30 s. Following the pre-training trial, the first trial started. At the beginning of each trial, the mouse was placed in the start point, the trial ended when the mouse enters the goal tunnel or after 3 min have elapsed, the mouse was allowed to stay in the tunnel for 1 min. Mice were trained four trials per day for 4 days. Trial interval was 15 min. After each trial the maze was cleaned with 5% ethanol solution. Trials were recorded and analyzed by EthoVision tracking system.

**Electrophysiological recordings**. Acute hippocampal slices (400 μm) were prepared by a vibratome (Vibratome, St. Louis, MO, USA) in ice-cold artificial cerebrospinal fluid(ACSF) containing 119 mM NaCl, 26.2 mM NaHCO$_3$, 2.5 mM KCl, 2.5 mM CaCl$_2$, 1.3 mM MgSO$_4$, 1 mM NaH$_2$PO$_4$ and 11 mM D-glucose, and recovered at 25–26 °C for at least 2 h before recording, saturated in 95% O$_2$ and 5% CO$_2$. For recording, slices were continuously perfused with O$_2$-saturated ACSF (25 °C) at a rate of 2 ml min$^{-1}$. Extracellular field potential were recorded by using glass electrodes (5–15 MΩ resistance, filled with 3 M NaCl) which was placed in stratum radiatum with ~100 μm away from stimulating electrode. Extracellular stimuli were administered on the border of the CA1 region along the Schaffer-collaterals every 15 s using bipolar platinum electrodes with 0.2 ms constant-current pulse.

All subsequent experimental stimulus strength was adjusted to obtain evoked extracellular field potential (fEPSP) with the initial slope of 0.15–0.20 mV ms$^{-1}$ in

**Fig. 6** Rapamycin treatment rescues electrophysiological deficits in *Phf8* null mice. **a** The slopes of fEPSP before and after tetanic stimulation are recorded from hippocampal slices. Despite that rapamycin administration does not affect the LTP induction and maintenance in wild-type mice (WT/Veh, n = 8 slices from six mice; WT/Rap, n = 7 slices from four mice; unpaired two-tailed *t*-test, p = 0.9317), the deficient LTP in *Phf8* null mice (KO/Veh, n = 8 slices from four mice) was rescued by treatment with low-dose rapamycin (KO/Rap, n = 8 slices from five mice; last 10 min of recording, unpaired two-tailed *t*-test, p = 0.0163). **b** The I-O curves show the relationship between fEPSP slope and stimulus intensity (two-way ANOVA, *F*(3, 436) = 7.132, p = 0.0001; Fisher's PLSD test, KO/Veh vs. KO/Rap, p = 0.0063). **c** The fiber volley amplitude measured at increasing stimulus intensities (two-way ANOVA, *F* (3, 436) = 5.987, p = 0.0005; Fisher's PLSD test, KO/Veh vs. KO/Rap, p = 0.0503). **d** The relationship between fEPSP slope and fiber volley amplitude. **e** Paired-pulse facilitation was similar among 4 groups (two-way ANOVA, *F* (3, 269) = 2.345, P = 0.732; Fisher's PLSD test, KO/Veh vs. KO/Rap, p = 0.2753). For the electrophysiological recording to evaluate basal transmission, we used 10 slices from seven mice for the WT group with vehicle injection and 10 slices from five mice for the other three groups. **f** Working model describing the mechanisms underlying the *Phf8* deletion induced cognitive impairment

each slice. Slices were stimulated for at least 30 min to establish a stable baseline before applying tetanus stimulus. Long-term potentiation (LTP) was induced with a tetanus stimulation (1 train of 100 Hz stimulation for 1 s). For the calculating of input–output relationships, the stimuli were delivered with gradually increasing current (from 10 μA to 100 μA in 10 μA increments). For the test of paired-pulse facilitation, two pulses with different inter-stimulus intervals (25, 50, 100, 200, or 300 ms) were applied.

**mRNA profiling**. Total RNA was isolated from hippocampi using TRIzol reagent (Invitrogen). cDNA was synthesized using reagents from a reverse transcription kit (FSQ 301, Toyobo). qPCR was performed using SYBR Green 5× PCR Master Mix (QPS-201, Toyobo) in a real-time PCR system (Applied Biosystems). All reactions were performed in triplicate in at least three independent experiments. The primer pair efficiencies were determined based on the slope of standard curve that was generated after plotting the results of the titration of the target cDNA ($C_T$ vs. Logarithm of template concentration). The efficiency is calculated as: $E = 10^{(-1/slope)} - 1$. The efficiency of primer pair should be between 90–110%. All the primer sequence and amplification efficiencies are listed in Supplementary Table 3 and Supplementary Table 4.

**Microarray data analysis**. In the microarray experiment, hippocampus from three *Phf8* KO mice or their wild littermates were tested as a sample for affymetrix chip. The data was analyzed by GeneSpring software, and filtered the probe with 'A' call besides 1.5-fold change cutoff criterion used in the differential expression. As none PHF8 antibodies were found appropriate for Chip sequence of mice tissue so far, we then downloaded PHF8 Chip-seq data from Helin's lab which were obtained from human neuroblastoma cells, all RefSeq genes with binding signals on promoter regions (TSS+−2kb) were identified as PHF8-bound proteins. Taking account of the orthologous genes in human and mouse, we converted the phf8-bound human gene names into mouse's homologous gene names (using the table of homologous genes of human and mouse reported previously[26]. These genes were chosen to combine with differentially expressed genes from microarray, and GO analysis was then conducted on DAVID website (https://david.ncifcrf.gov/).

**Immunohistochemistry**. Mice were anaesthetized with sodium pentobarbital and perfused transcardially with saline followed by 1% paraformaldehyde (PFA) with 1% sucrose in 0.1 M phosphate buffer (PB, pH 7.4). The brains were then removed and post-fixed in the same fixative for 4 h at 4 °C, and immersed in 30% sucrose in PB for 24–48 h at 4 °C Transverse brain sections (14 μm) were cut in a microtome cryostat and processed for immunofluorescence. All the sections were blocked with 5% normal goat serum in 0.01 M phosphate-buffered saline (PBS, pH 7.4) with 0.5% Triton-X-100 for 1 h at room temperature and incubated over 48 h at 4 °C with primary antibody PHF8 (ab36068, Abcam) and NeuN (MAB377, Millipore). The sections were then incubated for 2 h at room temperature with Alexa 488 conjugated goat anti rabbit IgG (1:500, Molecular Probes), Alexa 594 conjugated goat anti mouse IgG (1:1000, Molecular Probes), and DAPI (1:1000, Sigma). Omission of the primary antibody served as a negative control. The stained sections were examined with a laser scanning confocal microscope (TCS SP5, Leica).

**Western blotting**. The protein extracted from hippocampus was quantified using the DC protein assay (Bio-Rad). Equivalent protein were subjected to 8% SDS–PAGE, and transferred electrophoretically to nitrocellulose membranes. After blocking with 1% BSA, membranes were incubated with primary antibodies (from Cell Signaling Technology): RSK1 (8408, 1:1000), phospho-RSK1 (9346, 1:1000), S6 (2217, 1:1000), phosphorylated S6 (4858, 1:1000), and Kras (ab172949, 1:1000, abcam), CaMKII (PA5-35501, 1:500, Thermo), Phospho-CaMKII (AP20855PU, 1:500, Acris) mTOR (2983 S, 1:2000, Cell Signaling Technology), Phospho-mTOR (2971 S 1:2000, Cell Signaling Technology), β-Tubulin (T5201, 1:10,000, Sigma) or β-actin (a3854, 1:10,000; Sigma), GAPDH (M20006, 1:10,000, Abmart), then membranes were incubated with horseradish peroxidase-conjugated secondary antibody. The blots were detected using a chemiluminescence method (ECL system, GE Healthcare, Chalfont St Giles, UK) and exposed to radiography films. The intensity of blots was quantified with densitometry using Image J software (National Institutes of Health). The average blot density from the wild-type groups was set as 100%. The relative density values from *Phf8* null mice were determined by the average value of the wild-type groups after each was normalized to the corresponding β-actin. All uncropped blots are shown in Supplementary Fig. 9.

**Chromatin immunoprecipitation**. Chromatin was prepared from hippocampi of wild-type and *Phf8* KO mice. The tissue was broken up and sonicated to a range of 0.3 to 0.5 kb. ChIP-qPCR experiments for PHF8 samples were carried out using the PHF8 antibody (93313, Novus), H4K20me1 antibody (ab9051, Abcam) and H3K9me2 (ab1220, Abcam) in a buffer (20.0 mM Tris-HCL, 0.1% SDS, 1% Triton-X-100, 2 mM EDTA, 150 mM NaCl). The final ChIP DNA and input were dissolved in ddH₂O and used for real-time PCR. Primers used were shown in Supplemental Fig. 9. Several PHF8 antibodies were tried for Chip sequence assay, however, none is found adequate for sequence analysis after immunoprecipitation. H4K20me1 was performed as described before, and ChIP libraries were prepared with NEB ultra DNA ChIP Library Kit and sequenced on an Illumina HiSeq X.

**Rapamycin treatment**. We dissolved fresh rapamycin (LC Laboratories) in the vehicle solution (specified below) before use. For in vivo experiments, we administered rapamycin intraperitoneally once daily at a dose of 5 mg kg⁻¹ (vehicle: 5% Tween 80 and 5% PEG 400 in ddH₂O). We administered injections for 3 days before tests and on the day of the behavioral tests. Specifically, injections were performed 3 h before the contextual fear conditioning test and 3 h before beginning training in the water maze experiment. For the electrophysiology experiment, the mice were intraperitoneally injected with rapamycin (5 mg kg⁻¹) for 3 days before and on the day of the electrophysiological recording.

**Data availability**. Microarray and ChIP-seq data have been deposited in the Gene Expression Omnibus database under accession code GSE107292. The authors declare that all data supporting the findings of this study are available within the article and its Supplementary Information files or from the corresponding author on reasonable request.

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

## Acknowledgements

This work was supported by the National Major Scientific Instruments Development Project (2013YQ03092305), National Basic Research Program of China (2014CB943100), "Strategic Priority Research Program" of the Chinese Academy of Sciences (XDB19000000), the Major Research plan of the National Natural Science Foundation of China (91632103), the National Nature Science Foundation of China (31300895, 81421061, 81403463, 81361120389), the Shanghai Municipal Commission of Science and Technology Program (14JC1403700, 13DJ1400303, 13DZ2260500), Program of Shanghai Subject Chief Scientist (17XD1401700) "Eastern Scholar" project supported by Shanghai Municipal Education Commission, the fund of Shanghai Jiao Tong University (15JCZZ02).

## Author contributions

W.L., C.D.C. designed the experiments and wrote manuscript; X.C., Y.Z., and S.W. carried out the histochemical and behavioral experiments and wrote manuscript; Q.X. and A.P.C. carried out the behavioral experiments; Y.H. and S.L. carried out electrophysiology experiments; Y.D. and X.W. helped in the biochemical experiments; L.S., Q.X. assisted for animal breeding; Z.Z. and L.X. established the *Phf8* knockout mice; L.Y. analyzed the sequencing data; J.S., E.T., G.H., L.H. helped to analyze the data.

## Additional information

**Competing interests:** The authors declare no competing financial interests.

