## [Peer Review File · Nature Communications]

Reviewers' comments:

Reviewer #1 (Remarks to the Author):

phf8 is a candidate gene for XMLR. By generating a KO mouse for this genes, the authors were able to show that learning and memory are compromised in phf8 KO mice, based on Morris water maze and fear conditioning tests. Long term potentiation is also affected in phf8 KO mice as well as the basal synaptic transmission. Other experiments suggest that these effects are due to postsynaptic changes in phf8 KO mice, but not further investigated.

To identify the molecular mechanisms, the authors performed a microarray analysis in hippocampus and found about 2500 genes deregulated. A consistent portion of them are direct target of phf8 (evaluated by ChIP with specific antibodies against PHF8 protein performed in other studies). They found hyperactivation of mTOR signalling, with RSK and S6 hyper-phosphorylated. For the molecular mechanism, the authors analysed the presence of phf8 and the level of K9me2 and K20me1, know substrates of phf8 catalytic activity, at 3 targets genes involved in the mTOR pathway. The authors found that phf8 is specifically recruited at the TSS of these genes and that genetic ablation of phf8 results in increased H4K20me1 at TSS of the genes. Finally, the authors test the effect of mTOR activity inhibition in vivo, exposing KO mice to low doses of rapamicin and testing their behaviours and other relevant synaptic parameters.

Overall, the findings reported in the manuscript are interesting and novel. The in vivo analysis performed using rapamicin administration in KO mice suggests that the use of this drug might be beneficial in the treatment of XMLR cases. The statistical analysis is correctly reported in the figure legends and the manuscript is clearly written.

I strongly suggest that the authors include in the manuscript other tests for learning/memory, to strengthen their findings. These other in vivo tests may be also useful to confirm the rescue of the phenotypes obtained with rapamicin treatment, that is at the moment not fully convincing.

More specific points:

- 1) Representative swim trajectory of one of the trial should be inserted in Figure 1a.
- 2) What about adaptive or social behaviour tests, also common in disease? This should be at least discussed, in particular if tested with negative results.
- 3) A comprehensive analysis of both up and down regulated genes should be included, as phf8 targets the repressive H3K9 methylation mark, that should be increased in phf8KO mice theoretical leading to gene downregulation. Go Analysis of downregulated gens should be included in Figure 3.
- 4) In Figure 4, the authors need to include as control the H4K20me1 level in genes not deregulated in phf8 KO mice compared to wild type animals. It will be also relevant to test if the recruitment of PolII is enhanced at TSS of deregulated genes by performing ChIP. Is not required, but the authors should consider the possibility to perform H4K20me1 ChIP and deep sequencing to see the correlation with phf8 deregulated targets.
- 5) In figure S6 the authors show the % of freezing time is higher in WT/Veh compared to the animals treated with rapamicin. The authors should provide a possible explanation for this evidence.

Minor points:

- The images in Supplemental Fig. 1d need to be improved as they are not clearly visible.
- LTP (long term potentiation) should be defined in line 131 and not later (line 134).
- What is the dotted line represented in figure 3b? It should be specified in the legend.
- In figure 4b-c, I recommend to use "H3K9me2" or "H4K20me1" instead to K9-2 or K20-1 in the Y axes.

- In some cases, the figures are not described in detail in the legends. For example: the small black squares in fig 4a should be indicated in the legend, the graph in figure 3 d should be described in the legend.
- Molecular weight markers should be indicated in the Western blots presented.

Reviewer #2 (Remarks to the Author):

In this manuscript, Chen and colleagues investigated the role of Phf8, a histone demethylase, in neurodevelopment and cognition. PHF8 has been implicated in syndromal and non-specific forms of X-chromosome-linked intellectual disability (XLID). The authors show that Phf8 knockout mice display normal brain development but impaired learning and memory, and compromised long-term potentiation (LTP) in the hippocampus. Molecular analyses suggested that Phf8 could regulate the activity of mTOR signaling pathway. Although the manuscript is potentially interesting, however, the data presented in the manuscript are still quite preliminary and there are many gaps within the manuscript that need to be addressed, particularly the molecular analyses of Phf8 targets. Below are some of my specific comments:

1. The authors indicated that Phf8 KO mice display normal brain development, however, there was no data supporting this claim. Detailed morphological analyses of neurons in Phf8 KO mice are needed.
2. The authors demonstrated the learning and memory deficit in Phf8 KO mice using Morris water maze assay. Additional behavioral assays are needed to further support this conclusion.
3. It is unclear how the authors carried out the microarray experiment? Description of microarray analyses is completely missing in the manuscript. It seems that the criteria that they used to call differentially expressed (DE) genes was a fold change > 1.5 or < -1.5 . This is a very lax cut-off without convinced statistical analysis. It is difficult to evaluate the significance of RSK1 gene expression change in the absence of these descriptions.
4. To identify the direct targets of Phf8, the authors utilized a previously published Phf8 ChIP-seq data to cross-compare with microarray analyses result. The previously published PHF8 ChIP-seq data were generated using human neuroblastoma cells. The authors tried to compare between human ChIP-seq data and gene expression data generated from mouse. The authors need to generate mouse ChIP-seq data for this comparison.
5. The authors exclusively focused on RSKs-mTOR-S6K signaling cascade, however, it is unclear this pathway is the main pathway altered by the loss of Phf8. So extensive molecular analyses of Phf8 binding and its impact on gene expression are needed.

Reviewer #3 (Remarks to the Author):

In this manuscript Chen et al. generate a KO mouse for PHF8, a histone demethylase linked to intellectual disability. The PHF8 KO showed impaired spatial memory and LTP deficits. By microarray analysis, the authors found deregulation of about 2,500 genes in the PHF8 KO hippocampus, and focused their interest on RSK1, a kinase known to activate mTOR.

Based on the increased phosphorylation of S6 protein (a downstream target of mTOR) at Ser235/236, the authors conclude that mTOR is hyperactive, and protein translation is enhanced in PHF8 KO. However these points are just presumed, but not experimentally proven. Considering the complexity of the mTOR signaling pathway, other possibilities cannot be excluded. For example, RSK1 has been shown to phosphorylate S6 at residues Ser235/236 directly (Roux et al., "RAS/ERK signaling promotes site-specific ribosomal protein S6 phosphorylation via RSK and stimulates cap-dependent translation", JBC 2007).

The authors treated PHF8 KO mice with rapamycin, and claimed to recover spatial memory. However, this assertion is not clearly sustained by the data: Fig. 5a shows that vehicle-treated KO mice did not show memory deficits at probe test 3 (although untreated KO mice did, Fig. 1a). This puzzling result suggests that vehicle has a similar therapeutic effect than rapamycin. Another confusing result is the fact that WT animals treated with vehicle (Fig. 5a) clearly performed worse than untreated WT mice at probe test 1 (Fig. 1a). In fact, no difference is observed among the four experimental groups at probe test 1 (Fig. 5a), which visibly contrasts with the data shown in Fig. 1a.

One more unclear point is related to the effect of rapamycin on LTP recovering. Was rapamycin intraperitoneally administered as indicated in the main text, or was it added to the slices during electrophysiological recordings, as stated in the Method section? Why rapamycin did not impair LTP in WT slices, as originally showed by Tang et al. ("A rapamycin-sensitive signaling pathway contributes to long-term synaptic plasticity in the hippocampus", PNAS, 2002)?

Related to the novelty of the results, a relevant paper has been inexplicably omitted: the report by Liu et al. showing that PHF8 inhibits the Akt-mTOR pathway ("The histone demethylase PHF8 represses cardiac hypertrophy upon pressure overload", Exp Cell Res, 2015). It is also worth mentioning that, in the Methods section, authors describe the generation of embryonic stem cells disrupted for PHF8, although it has been previously published (Tang et al., "Plant Homeo Domain Finger Protein 8 Regulates Mesodermal and Cardiac Differentiation of Embryonic Stem Cells Through Mediating the Histone Demethylation of pmaip1", Stem Cells, 2016). Thus, panel "a" of Fig S1 in this manuscript is almost identical to Fig. S1A of the Stem Cells paper.

The text requires extensive revision. Figure legends lack of important information in some cases (number of replicates, axe units...) or is probably erroneous (for example, legend of Fig. S1, panel b). Regarding the Methods, important technical details that are critical for the reliability of the results are omitted. In particular, the authors should provide information about the efficacy of the primers used in quantitative PCR, and how they assessed the linearity of the antibodies used in quantitative western-blot. Microarray analysis is not described in the Methods. Finally, the list of deregulated genes identified by microarray should be included as a supplemental file.

We have studied comments carefully and have made corrections which we hope meet with approval. Revised portions are marked in red in the manuscript. The main corrections in the paper and the responses to the reviewers' comments are as follows:

Reviewer#1

Specific points:

I strongly suggest that the authors include in the manuscript other tests for learning/memory, to strengthen their findings. These other *in vivo* tests may be also useful to confirm the rescue of the phenotypes obtained with rapamycin treatment, that is at the moment not fully convincing.

Answer: Both reviewer #1 and reviewer #2 suggested us to do additional behavioral assays. Other learning and memory tests including passive avoidance performance and Barnes maze were added in Figure 5b-5d. KO mice displayed the learning and memory deficits in these tasks and the impairments were rescued by rapamycin treatment. Thanks for the suggestion and the additional data make the conclusion more convincing.

1. Representative swim trajectory of one of the trials should be inserted in Figure 1a.

Answer: According to reviewer's suggestion, we inserted swim trajectory in Figure 1a and also in Figure 5a.

2. What about adaptive or social behaviour tests, also common in disease? This should be at least discussed, in particular if tested with negative results.

Answer: In the introduction, we mentioned "Intellectual disability is a heterogeneous neurodevelopmental disorder characterized by impaired intellectual and adaptive functioning". Adaptive function was engaged in individual's daily living skills, including Conceptual Reasoning, Social Interactions, and Practical Functioning in human. Patients with PHF8 mutation are characterized by mental retardation with or without cleft lip/cleft palate. (AM Koivisto et al. Clin Genet. 2007). Although we focus on the role of PHF8 in regulating learning and memory in this manuscript, according to reviewer's comment, we did the social behavioral test and find no significant difference between the Phf8 KO mice and their wild type littermates (Figure S6 d). We also added the sentences in the "Discussion" as follows, "Beyond intellectual disability, impaired adaptive functioning is also observed in XLID patients. Social communication tests and behaviors in response to the environment are used to evaluate adaptive functions in mouse model studies, Phf8 KO mice showed no significant difference in the social ability test here. However, Phf8 KO mice display hyperactivity in the unfamiliar environment of open field and impairments in passive avoidance test and fear conditioning. In these unavoidable and unexpected conditions, mice need adaptive functions to deal with the stress and tasks. The role of PHF8 in adaptive behavior requires further investigations for making conclusions as we did for learning and memory here."

3. A comprehensive analysis of both up and down regulated genes should be included, as phf8 targets the repressive H3K9 methylation mark, that should be increased in phf8KO mice theoretically leading to gene downregulation. GO Analysis of downregulated genes should be included in Figure 3.

Answer: We included GO Analysis of downregulated genes in Figure 3a and also added a list of upregulated and downregulated genes in supplementary table.

4. In Figure 4, the authors need to include as control the H4K20me1 level in genes not deregulated in phf8 KO mice compared to wild type animals. It will be also relevant to test if the recruitment of PolII is enhanced at TSS of deregulated genes by performing ChIP. Is not required, but the authors should consider the possibility to perform H4K20me1 ChIP and deep sequencing to see the correlation with phf8 deregulated targets.

Answer: We added ChIP-qPCR results of *Mapk14* in Figure 4. *Mapk14* is one of PHF8 binding genes, and we confirmed that *Mapk14* mRNA expression was not changed in Phf8 KO mice. ChIP-qPCR results showed that PHF8 bound at the TSS of *Mapk14*, however, the enrichment of both H3K9me2 or H4K20me1 were not changed at the TSS of *Mapk14*.

In our current manuscript, we focused more on genes as *Kras*, *Camk2d* and *Rps6ka1* in the

LTP pathway, thus, we performed Chip-qPCR to confirm that H4K20me1 contributed to the regulation of these genes. We agree that H4K20me1 CHIP sequencing will be of great help in deep understanding of its correlation with all other PHF8 regulated targets, and we will conduct deep sequencing experiments in the future.

5. In figure S6 the authors show the % of freezing time is higher in WT/Veh compared to the animals treated with rapamycin. The authors should provide a possible explanation for this evidence.

Answer: Rapamycin, the mTOR inhibitor, is a regulator of mRNA translation and considered to be involved in various forms of synaptic plasticity and memory consolidation. (Parsons RG et.al, Translational control via the mammalian target of rapamycin pathway is critical for the formation and stability of long-term fear memory in amygdala neurons. *J Neurosci.* 2006). However, it was also reported that WT rats or mice with rapamycin treatment decreased the percentage of freezing compared to WT/Veh. (J Blundell et.al Systemic inhibition of mammalian target of rapamycin inhibits fear memory reconsolidation. *Neurobiology of Learning and Memory*, 2008). This result is consistent with our data that WT/Veh mice displayed higher freezing percentage compare to mice treated with rapamycin, a possible explanation is that systemic inhibition of the mTOR pathway by rapamycin might reduce contextual fear memory consolidation related to amygdala functions.

Minor points:

- The images in Supplemental Fig. 1d need to be improved as they are not clearly visible.

Answer: According to the upload file size limits, we used a low-resolution image in Fig S1 d. And this time, the pixel of images was improved.

- LTP (long term potentiation) should be defined in line 131 and not later (line 134).

Answer: The definition of LTP was given in first sentence of this paragraph.

- What is the dotted line represented in figure 3b? It should be specified in the legend.

Answer: The dotted line indicated that 1.5-fold change in mRNA level of knockout mice compared to wild type mice was used as a threshold.

- In figure 4b-c, I recommend to use “H3K9me2” or “H4K20me1” instead to K9-2 or K20-1 in the Y axes.

Answer: We changed the headline of Y axes according to the recommendation.

- In some cases, the figures are not described in detail in the legends. For example: the small black squares in fig 4a should be indicated in the legend, the graph in figure 3d should be described in the legend.

Answer: Thanks for pointing out these shortcomings. We carefully checked and revised them in figure legends.

- Molecular weight markers should be indicated in the Western blots presented.

Answer: Molecular weight was added in Figure3c,3d and Figure S1b and Figure S6.

Reviewer #2

1. The authors indicated that Phf8 KO mice display normal brain development, however, there was no data supporting this claim. Detailed morphological analyses of neurons in Phf8 KO mice are needed.

Answer: Thanks for the reviewer’s comments. In fact, we checked the brain size and gross morphology of neurons, and there was no obvious differences between the Phf8 KO and WT. In that case, that’s really inappropriate to conclude that Phf8 KO mice display normal brain development, and we changed the words to “Here we report that PHF8 knockout mice displayed impaired learning and memory and compromised long-term potentiation (LTP), whereas had no gross morphological defects” in “Abstract”.

2. The authors demonstrated the learning and memory deficit in Phf8 KO mice using Morris water maze assay. Additional behavioral assays are needed to further support this conclusion.

Answer: We further performed learning and memory behavior tests including passive avoidance performance and Barnes maze tests, and the data were shown in Figure 5b-5d, which support the conclusion of memory deficits in KO.

3. It is unclear how the authors carried out the microarray experiment? Description of microarray analyses is completely missing in the manuscript. It seems that the criteria that they used to call differentially expressed (DE) genes was a fold change > 1.5 or < -1.5 . This is a very lax cut-off without convinced statistical analysis. It is difficult to evaluate the significance of RSK1 gene expression change in the absence of these descriptions.

Answer: We included the Microarray data analysis in methods section. We also think that 1.5 fold change as a cut-off is quite loose, and we set the criteria of fold change more rigorous, however, a small number of genes were found with a fold change >2 or <-2 and insufficient for the GO analysis of pathways. Thus, a 1.5-fold change criteria was applied to our microarray analysis, and the long-term potential pathway was shown in GO analysis. However, our consequent qPCR and western blot experiments confirmed that these LTP associated genes were really overexpressed in the mRNA and protein level.

4. To identify the direct targets of Phf8, the authors utilized a previously published Phf8 ChIP-seq data to cross-compare with microarray analyses result. The previously published PHF8 ChIP-seq data were generated using human neuroblastoma cells. The authors tried to compare between human ChIP-seq data and gene expression data generated from mouse. The authors need to generate mouse ChIP-seq data for this comparison.

Answer: We tried our best to perform Phf8 CHIP sequence in many ways for a few years. However, both commercial and customer-made PHF8 antibodies were not satisfied for Chip-seq in mice. We tried antibodies Ab36068 from Abcam, 93313 and 93314 from Novus, and another antibody produced by ourselves which is done well for Chip in 293T cell, all of them are adequate for western blot and IP in mice (shown in below fig a, upper PHF8 IP; lower, IgG as a control). Furthermore, we sonicated samples from mice hippocampus into a Chip-seq required DNA fragments like below (about 200-300bp, shown in fig b), neither of the antibodies works. For immunoprecipitation assay, we applied various antibodies with or without sonication (shown in fig c). We conducted different cross-linking time courses, several different sonication buffers with different sonication protocols, and other methods like enzyme digestion to get the DNA fragments required in CHIP sequence (enrich at 100-300bp), none of the antibodies works well in the further IP experiments of mice tissue. Then, we comprised in the size of DNA fragments and treated the samples from hippocampus with less sonication, in which, the DNA fragments were assembled at about 500bp (as shown in fig d lane 4), the antibody 93313 from Nuvus works, which made it possible for us to perform the Chip-qPCR experiments.

As in the lack of PHF8 Chip-seq data of mice, we downloaded data of human neuroblastoma cells from Helin's lab, all promoter regions (TSS \pm 2kb) with binding signal of the RefSeq gene were identified as PHF8-bound proteins. We assessed orthologous splicing isoforms in human and mouse orthologous genes, and converted these human homologous gene names into mouse (using a table of homologous genes of human and mouse reported previously: Assessment of orthologous splicing isoforms in human and mouse orthologous genes, BMC Genomics 2010, 11:534). Then these genes were overlapped with differentially expressed genes in microarray analysis.

5. The authors exclusively focused on RSKs-mTOR-S6K signaling cascade, however, it is unclear this pathway is the main pathway altered by the loss of Pfh8. So extensive molecular analyses of Pfh8 binding and its impact on gene expression are needed.

Answer: According to our microarray and ChIP data analysis, several pathways were regulated by the deletion of Pfh8. As we know, Pfh8 is a candidate gene for X-chromosome-linked intellectual disability. We were more interested in the Pfh8 function on learning memory and firstly focused on the LTP pathway. Furthermore, we confirmed that RSKs-mTOR-S6K signaling was involved in the memory deficit in Pfh8 KO mice. We also believe that the Pfh8 could play very important and broad roles on other pathways and biological functions. This is our first attempt and effort to find one of the key pathways which contribute to the XLID from our Pfh8 KO mice. We would like to share the finding with the colleagues in the field and continue the extensive works on the Pfh8 function in brain as the review's wishes and suggestion.

Reviewer#3

1. Based on the increased phosphorylation of S6 protein (a downstream target of mTOR) at Ser235/236, the authors conclude that mTOR is hyperactive, and protein translation is enhanced in Pfh8 KO. However these points are just presumed, but not experimentally proven. Considering the complexity of the mTOR signaling pathway, other possibilities cannot be excluded. For example, RSK1 has been shown to phosphorylate S6 at residues Ser235/236 directly (Roux et al., "RAS/ERK signaling promotes site-specific ribosomal protein S6 phosphorylation via RSK and stimulates cap-dependent translation", JBC 2007).

Answer: Thanks for review's comment. We agreed that mTOR signal pathway is complex. Besides RSK1 directly phosphorylate S6, other studies have shown that RSK-mediated phosphorylation of Raptor is important for mTOR complex1 activation (A. Carrière, et. al Curr. Biol., 18 (2008), p. 1269). Thus, ribosomal protein S6 phosphorylation could be regulated by mTOR signaling pathway or RSK1 directly as the reviewer said. Our data showed that increased phosphorylation of S6 protein in Pfh8 KO mice and could be reduced by low-concentration rapamycin treatment. Furthermore, Pfh8 KO mice showed improve LTP in electrophysiological recording as well as behavior test such as water maze, passive avoidance and Barnes maze after rapamycin treatment. We couldn't exclude the possibility that RSK1 directly phosphorylate S6 as reviewer said so we changed the mTOR is hyperactive as "mTOR signaling is hyperactive"

2. The authors treated PHF8 KO mice with rapamycin, and claimed to recover spatial memory. However, this assertion is not clearly sustained by the data: Fig. 5a shows that vehicle-treated KO mice did not show memory deficits at probe test 3 (although untreated KO mice did, Fig. 1a). This puzzling result suggests that vehicle has a similar therapeutic effect than rapamycin. Another confusing result is the fact that WT animals treated with vehicle (Fig. 5a) clearly performed worse than untreated WT mice at probe test 1 (Fig. 1a). In fact, no difference is observed among the four experimental groups at probe test 1 (Fig. 5a), which visibly contrasts with the data shown in Fig. 1a.

Answer: Thanks for review's comments. Morris water maze was widely used to test spatial learning and memory in rodent, however it was influenced by many factors, such as training procedure, animal age and stress.

In our experiments, we thought that the mutant mice displayed "slower learner" in Water maze test. Actually, we tried water maze test for several times and confirmed learning and memory deficits in naïve Phf8 KO mice comparing to naïve WT all the time. Sometime the whole groups of animals learned fast sometime slow, so we always do water maze with the complete groups of enough animal number (around 15) per group. And the result of water maze was repeated three times, one time KO mice display normal learning and memory in Probe test 3(See figure below). It suggested that KO mice might learn this task after 7 days training and might not completely lose the learning skill.

Fig: WT, n=12; KO, n=11; paired two-tailed t-test for water maze test (* $p < 0.05$). Four quadrants: adjacent left (AL), target quadrant (T), adjacent right (AR), opposite quadrant (O)

With the rescue experiment, we presumed that daily injections might affect the performance of mice. The data (Fig 5a) indicated that KO mice have obvious learning deficit compared to WT mice on probe test 2, and the rapamycin injection indeed rescued the learning deficit. After 7 day training the Phf8 KO mice learned the location of the platform without Rapamycin treated, which indicated the Phf8 KO mice could learn with injection protocol. The data were also repeatedly confirmed three times.

3. One more unclear point is related to the effect of rapamycin on LTP recovering. Was rapamycin intraperitoneally administered as indicated in the main text, or was it added to the slices during electrophysiological recordings, as stated in the Method section? Why rapamycin did not impair LTP in WT slices, as originally showed by Tang et al. ("A rapamycin-sensitive signaling pathway contributes to long-term synaptic plasticity in the hippocampus", PNAS, 2002)?

Answer: Rapamycin was daily administered to mice by intraperitoneal injection 3 days before electrophysiological experiment, and we didn't use rapamycin perfusion in slice throughout the recording. We have corrected the mistakes in the manuscript. In Tang's paper, rapamycin was applied to brain slice in ACSF at least 30 min before the first tetanus and inhibited the magnitude of late-phase LTP (230-240 min after the last tetanus, whereas the early phase of LTP (50-60 min after the last tetanus) was unaffected. According to our result, rapamycin didn't impair in early phase of LTP which was consistent with conclusion in that paper.

4. Related to the novelty of the results, a relevant paper has been inexplicably omitted: the report by Liu et al. showing that PHF8 inhibits the Akt-mTOR pathway (“The histone demethylase PHF8 represses cardiac hypertrophy upon pressure overload”, *Exp Cell Res*, 2015). It is also worth mentioning that, in the Methods section, authors describe the generation of embryonic stem cells disrupted for PHF8, although it has been previously published (Tang et al., “Plant Homeo Domain Finger Protein 8 Regulates Mesodermal and Cardiac Differentiation of Embryonic Stem Cells Through Mediating the Histone Demethylation of pmaip1”, *Stem Cells*, 2016). Thus, panel “a” of Fig S1 in this manuscript is almost identical to Fig. S1A of the Stem Cells paper.

Answer: Thanks a lot for the review’s comment. The two papers focused on the function of PHF8 in cardiac system. One of our corresponding authors - Charlie Degui Chen - made the phf8 KO mouse. In our paper, we worked on the learning memory mechanism of phf8 KO mice. Dr.Chen also collaborated with Dr.Huang-Tian Yang as the co-author of the Stem Cells paper to work on the function of PHF8 in cardiac system. The strategy for the design of targeted disruption of the catalytic JmjC domain of PHF8 in our mice is the same as it’s in embryonic stem cell. We deleted the panel “a” of Fig S1 and cited the papers in our manuscript. In the discussion, we cited the publication as reviewer suggestion. “Besides, Akt-mTOR pathway was reported to be regulated by PHF8 in cardiac system and rapamycin treatment could rescue the effects of PHF8 loss in neonatal rat ventricle myocytes”

5. The text requires extensive revision. Figure legends lack of important information in some cases (number of replicates, axe units...) or is probably erroneous (for example, legend of Fig. S1, panel b). Regarding the Methods, important technical details that are critical for the reliability of the results are omitted. In particular, the authors should provide information about the efficacy of the primers used in quantitative PCR, and how they assessed the linearity of the antibodies used in quantitative western-blot. Microarray analysis is not described in the Methods. Finally, the list of deregulated genes identified by microarray should be included as a supplemental file.

Answer: We amended the figure legends carefully in this revision version based on the reviewer’s suggestion. Substrates gradient dilution was used to evaluate the efficacy of the primers and gradient dilution of protein samples were used to evaluate the linearity of the antibodies. We added Microarray analysis description in the Methods. We made the excel file for the differentially expressed gene of microarray in supplementary table.

Reviewers' comments:

Reviewer #1 (Remarks to the Author):

The authors addressed my suggestions and criticisms and the revised version of the manuscript of Chen et al, is, in my opinion, improved. In particular, the authors performed novel behavioural tests, as also suggested by a second reviewer. Importantly, the phenotypes tested were rescued by Rapamycin treatment, thus reinforcing the main message of the manuscript.

I only have 2 minor comments:

I noticed a discrepancy between the text (line 163) and figure 3a, in the number of gene deregulated that should be corrected. I also suggest to add in M&M a note regarding the limitations of the PHF8 antibodies in ChIP/sequencing.

Reviewer #2 (Remarks to the Author):

In this revised manuscript, Chen and colleagues have addressed many questions raised by the reviewers during the first round of review. However, there are still two remaining important issues that the authors need to address:

1. Microarray experiment-In the revised manuscript, the authors included the detailed description of the experiment. They indicated, "In the microarray experiment, hippocampus from six PHF8 knock out mice or their wild littermates were mixed together as a sample for affymetrix chip." Essentially they only ran one pair of samples. Typically for the microarray experiments, one would need triplicate for each condition. What they have in the manuscript is unacceptable. Figure 3 could be very misleading.

2. PHF8 ChIP experiments-I do understand the potential issues. However, it seems that the authors have solved the problems by sonicating the chromatin into ~500 bp fragments. Their ChIP-PCR seems pretty clean. Even with larger DNA fragments, the authors should be able to generate the ChIP-seq library and identify PHF8 binding sites. The resolution might not be as good as smaller fragments, but the data will be very informative for the present study. In addition, the authors should include H4K20me1 CHIP-seq data to strengthen their conclusions as suggested by the other reviewer.

Reviewer #3 (Remarks to the Author):

Some concerns previously raised need further clarification.

- Point 1 has not been experimentally addressed. The authors claimed that "Indeed, loss of PHF8 in the brain conspicuously increased phosphorylated RSK1 and S6 but not total expression of S6 (Fig. 3c and 3d), indicating hyperactive mTOR signaling and enhanced protein translation in mutant mice". I have to insist that hyperactive mTOR signaling, although probable, is not demonstrated. Increase in protein translation is also assumed by the authors (and I agree that it is likely) but has not been demonstrated. Thus, in the Discussion section, the authors cannot conclude: "Our results showed upregulation of Ras and RSK1 in PHF8-null hippocampus, led to overactivation of mTOR signaling and excessive protein translation"

- The answer to point 2 is not fully satisfactory. In the rebuttal letter, authors interpret the results shown in Figure 5a as KO mice learning slower than WT. However, these results correspond to the "memory test" phase of the water Morris maze. Since the authors do not show the performance of the different experimental groups during the training (i.e., learning) phase, the results and the

corresponding conclusions are still unclear. The authors should explain this important point in the text and discuss their interpretation appropriately.

- Point 5: Still some errors (for example, β -actin instead of β -actin). On the other hand, if the linearity of the signals in western blot has been experimentally proven, and the efficacies of primers used in real-time PCR are 100% (or other values, but these figures have been taken into account to perform the corresponding calculations), this information should be clearly mentioned in the Methods section.

We thank the reviewers for their wonderful comments. According to their invaluable points we revised the manuscript carefully with additional experiments and figures. Revised portions are marked in red in the manuscript. We also responded point by point to the reviewers' comments here. We hope that this revised manuscript is sufficient to meet the quality and scope of Nature Communications.

Reviewer #1 (Remarks to the Author):

"The authors addressed my suggestions and criticisms and the revised version of the manuscript of Chen et al. is, in my opinion, improved. In particular, the authors performed novel behavioural tests, as also suggested by a second reviewer. Importantly, the phenotypes tested were rescued by Rapamycin treatment, thus reinforcing the main message of the manuscript.

I only have 2 minor comments:

I noticed a discrepancy between the text (line 163) and figure 3a, in the number of genes deregulated that should be corrected. I also suggest to add in M&M a note regarding the limitations of the PHF8 antibodies in ChIP/sequencing."

Response: We appreciate the reviewer's positive opinion on the revised manuscript version. We also responded to the 2 minor comments below.

"I noticed a discrepancy between the text (line 163) and figure 3a, in the number of genes deregulated that should be corrected."

We redid the microarray experiments using 3 vs 3 mice in each group and have modified the results in main text, Figure 3 and Supplemental Figure 4.

"I also suggest to add in M&M a note regarding the limitations of the PHF8 antibodies in ChIP/sequencing."

The limitations of PHF8 antibodies in ChIP sequencing were added in the first paragraph of Microarray data analysis in Methods. "As none PHF8 antibodies were found appropriate for ChIP sequence of mice tissue so far,....."

Reviewer #2 (Remarks to the Author):

"In this revised manuscript, Chen and colleagues have addressed many questions raised by the reviewers during the first round of review. However, there are still two remaining important issues that the authors need to address:

1. *Microarray experiment - In the revised manuscript, the authors included the detailed description of the experiment. They indicated, "In the microarray experiment, hippocampus from six PHF8 knock out mice or their wild littermates were mixed together as a sample for Affymetrix chip." Essentially they only ran one pair of samples. Typically for the microarray experiments, one would need triplicate for each condition. What they have in the manuscript is unacceptable. Figure 3 could be very misleading."*

Response: We accepted the reviewer's point and redid the Microarray experiment by triplicated samples for each group. GO analysis showed that 1954 genes were over 1.5 folds up-regulated in KO mice, among which 526 were related to PHF8. We found that *Rps6ka1*, *Kras* and *Camk2d* were still upregulated more than 1.5 fold in Microarray assay and involved in both Neurotrophin and LTP pathway. We amended the

text, Figure 3 and Supplemental Figure 4. “As compared with control mice tissues, we found that 1954 genes were upregulated while 1581 genes were downregulated with at least 1.5-fold change in PHF8-deficient hippocampus..... Notably, *Rps6ka1*, *Kras* and *CamKII*, the key molecules in both Neurotrophin signaling pathway and LTP signaling pathway, were identified in gene ontology analysis in PHF8-deficient hippocampus (Supplemental Fig. 4a and 4b).”

2. PHF8 ChIP experiments-I do understand the potential issues. However, it seems that the authors have solved the problems by sonicating the chromatin into ~500 bp fragments. Their ChIP-PCR seems pretty clean. Even with larger DNA fragments, the authors should be able to generate the ChIP-seq library and identify PHF8 binding sites. The resolution might not be as good as smaller fragments, but the data will be very informative for the present study. In addition, the authors should include H4K20me1 CHIP-seq data to strengthen their conclusions as suggested by the other reviewer.

Response: According reviewer's concern, we did try several times of PHF8 chip experiments again, even using larger fragments. However, the DNA samples could not meet the quality requirement for sequence assay. Recently, other group published one paper associated with PHF8 knock out mice in Nature Communication (2017 May 9;8:15142.). They also performed PHF8 Chip-qPCR data of mice neocortex instead of sequence, so we speculated that it might be also due to the inefficiency of PHF8 antibody. The limitations of PHF8 antibodies in Chip sequencing were added in first paragraph of Microarray data analysis in Methods.

As the reviewer suggested, we added H4K20me1 CHIP-seq data in supplemental figure 7. As PHF8 binds the TSS region of genes and regulates transcription, we analyzed the TSS \pm 2kb region. The results showed increased H4K20me1 binding at the TSS of *Kras*, *Camk2d* and *Rps6ka1*, which is in accordance with the previous Chip-qPCR results.

Reviewer #3 (Remarks to the Author):

Some concerns previously raised need further clarification.

“Point 1 has not been experimentally addressed. The authors claimed that “Indeed, loss of PHF8 in the brain conspicuously increased phosphorylated RSK1 and S6 but not total expression of S6 (Fig. 3c and 3d), indicating hyperactive mTOR signaling and enhanced protein translation in mutant mice”. I have to insist that hyperactive mTOR signaling, although probable, is not demonstrated. Increase in protein translation is also assumed by the authors (and I agree that it is likely) but has not been demonstrated. Thus, in the Discussion section, the authors cannot conclude: “Our results showed upregulation of Ras and RSK1 in PHF8-null hippocampus, led to overactivation of mTOR signaling and excessive protein translation” “

Response: Thanks for reviewer's points. In addition to the data of increased phosphorylated RSK1 and S6, we added the mTOR and phos-mTOR protein western blotting experiments. The data reinforced the conclusion of hyperactive mTOR signaling in the PHF8 KO mice. However, we agreed that we still could not conclude “excessive protein translation”. We rewrote the discussion part to “Previous studies showed that disruption in mTOR pathway impairs learning/memory and synaptic plasticity^{20,33,34,36}. Our results showed hyperactive mTOR signaling in PHF8-mutant mice. And pharmacological inhibition of hyperactive mTOR signaling with rapamycin rescued the impaired LTP and memory loss in PHF8-mutant mice. Taking

together, our data indicated that PHF8-mediated overactivation of mTOR signaling is essential for normal cognition and synaptic plasticity.” We also added the data to supplemental figure 5 and modified the text in the result: “Thus, we examined the phosphorylation level of RSK1, mTOR and ribosomal protein S6 by immunoblotting. Indeed, loss of PHF8 in the brain conspicuously increased phosphorylated RSK1, mTOR and S6 but not total expression of mTOR and S6 (Fig. 3b and 3c, Supplemental Fig. 5a and 5b), indicating hyperactive mTOR signaling in mutant mice.”

- **Supplemental Fig. 5a and 5b**

“The answer to point 2 is not fully satisfactory. In the rebuttal letter, authors interpret the results shown in Figure 5a as KO mice learning slower than WT. However, these results correspond to the “memory test” phase of the water Morris maze. Since the authors do not show the performance of the different experimental groups during the training (i.e., learning) phase, the results and the corresponding conclusions are still unclear. The authors should explain this important point in the text and discuss their interpretation appropriately.”

Response: We added the data of the latency to acquisition hidden platform and average speed during the training phase in supplemental figure 8a-8b. With the rescue experiment, we presumed that daily I.P. injections as a strong stimulus might affect the water maze performance of mice. Unfortunately, the mechanisms are still unclear. We may investigate the puzzle in the future study. We added some sentences in the discussion:

“Morris water maze was widely used to evaluate spatial learning and memory in rodent. Our data (Fig 5a) indicated that vehicle-treated KO mice still displayed obvious learning deficit compared to vehicle-treated WT mice on probe test 2, and the rapamycin injection could rescue the learning deficit. However, we noticed that vehicle-treated KO and vehicle-treated WT mice did not show the exact same performance at probe test 1 and probe test 3 as those of naïve KO and WT mice in Fig 1a. It suggested that there might be some unknown effects on water maze performance by daily injection itself. The difference on water maze performance of mutant mice with or without daily injection was also seen in the other paper²⁰. The mechanism of the puzzle needs to be investigated in the future study.”

- **Point 5:** Still some errors (for example, β -acitn instead of β -actin). On the other hand, if the linearity of the signals in western blot has been experimentally proven, and the efficacies of primers used in real-time PCR are 100% (or other values, but these figures have been taken into account to perform the corresponding calculations), this information should be clearly mentioned in the Methods section.

Response: “ β -acitn” in figure 3c, 3d and figure S1b has been corrected to “ β -actin”. In addition, we add the following description in Methods section: The primer pair efficiencies were determined based on the

slope of standard curve that was generated after plotting the results of the titration of the target cDNA (C_T vs. Logarithm of template concentration). The efficiency is calculated as: $E = 10^{(-1/\text{slope})} - 1$. The efficiency of primer pair should be between 90-110%. All the primer amplification efficiencies are listed in Supplemental Fig 9 and Supplemental Fig 10

In addition to the response above, we also would like to discuss the issue below.

In the process of our revised manuscript, Ryan and his colleagues published the paper on Nature Communications recently and reported that *Phf8*-deficient mice showed resistance to depression-like and anxiety-like behaviors in mice and no intellectual disability. But our *Phf8* KO mice display a significant intellectual disability in water maze, fear conditioning, Barnes maze and passive avoidance performance. The difference of behavior phenotype in *Phf8*-deficient mice in two papers might due to different genetic background and behavior test protocol. In comparison with Ryan's behavioral tests, some protocols we used were different, for example, the fear-conditioning test protocol we used is standard protocol: a 5-minutes training with 3 foot shocks and a 5-minutes testing 24 hour later. However, in Ryan's protocol: a 7-minutes training session and a 3-minutes testing. In the fear-conditioning test, freezing level generally becomes more significant after first 1 or 2 min initial memory recall. (Valerio Rizzo et al. *Biological Psychiatry*,2017 Figures in below). Genetic background of mice should be also considered, since genetic background could affect the behavioral performance dramatically (Harry J. Han, *Strain Background Influences Neurotoxicity and Behavioral Abnormalities in Mice Expressing the Tetracycline Transactivator*, *The Journal of Neuroscience*,2012). Our mice were at least the fifth-generation progeny of a backcross into C57BL/6J mice while their mice maintained on a 129/B6 background.

Valerio Rizzo et al. *Biological Psychiatry*,2017

REVIEWERS' COMMENTS:

Reviewer #2 (Remarks to the Author):

The authors have addressed my concerns.

Reviewer #3 (Remarks to the Author):

- Point 2 is still problematic. From the new data included in Fig. S8 is clear that learning is not affected in any of the four experimental groups. Consequently, Fig. 5a remains a puzzle for me. Taking into account the Walsh et al. paper, that reports no signs of cognitive impairment in Phf8 KO mice, I consider this puzzle should be solved. Perhaps the analysis of the quadrant preference in the Barnes maze, which is the classical output of this test (rather than the number hole pokes shown in Fig. 5c), could shed some light on it.

- Point 5 still need some addressing. The authors explains now correctly the linearity controls made in the case of quantitative PCR, but no answer was given about the linearity of signals in western blots. I have to insist in this point, since it is a mandatory control in quantitative western blot.